# Silencing of the Chitin Synthase Gene Is Lethal to the Asian Citrus Psyllid, *Diaphorina citri*

**DOI:** 10.3390/ijms20153734

**Published:** 2019-07-31

**Authors:** Zhan-Jun Lu, Yu-Ling Huang, Hai-Zhong Yu, Ning-Yan Li, Yan-Xin Xie, Qin Zhang, Xiang-Dong Zeng, Hao Hu, Ai-Jun Huang, Long Yi, Hua-Nan Su

**Affiliations:** 1College of Life Sciences, Gannan Normal University, Ganzhou 341000, China; 2National Navel Orange Engineering Research Center, Ganzhou 341000, China; 3Saskatoon Research Center, Agriculture and Agri-Food Canada, Saskatoon, SK S7N 0X2, Canada

**Keywords:** *Diaphorina citri*, chitin synthase, diflubenzuron, RNA interference

## Abstract

Chitin synthase is a critical enzyme that catalyzes *N*-acetylglucosamine to form chitin, which plays an important role in the growth and development of insects. In this study, we identified a chitin synthase gene (*CHS*) with a complete open reading frame (ORF) of 3180 bp from the genome database of *Diaphorina citri*, encoding a protein of 1059 amino acid residues with the appropriate signature motifs (EDR and QRRRW). Reverse transcription-quantitative PCR (RT-qPCR) analysis suggested that *D. citri CHS* (*DcCHS*) was expressed throughout all developmental stages and all tissues. *DcCHS* had the highest expression level in the integument and fifth-instar nymph stage. Furthermore, the effects of diflubenzuron (DFB) on *D. citri* mortality and *DcCHS* expression level were investigated using fifth-instar nymph through leaf dip bioassay, and the results revealed that the nymph exposed to DFB had the highest mortality compared with control group (Triton-100). Silencing of *DcCHS* by RNA interference resulted in malformed phenotypes and increased mortality with decreased molting rate. In addition, transmission electron microscopy (TEM) and fluorescence in situ hybridization (FISH) also revealed corresponding ultrastructural defects. Our results suggest that *DcCHS* might play an important role in the development of *D. citri* and can be used as a potential target for psyllid control.

## 1. Introduction

The Asian citrus psyllid (ACP), *Diaphorina citri* Kuwayama (Hemiptera: Liviidae), is a notorious pest across Asia, USA, and Brazil, causing severe economic losses in the citrus industry [1]. However, the greatest threat of ACP arises from the transmission of “*Candidatus* Liberibacter asiaticus (*C*Las)”, which causes citrus greening disease, also called huanglongbing (HLB) [2]. Up to this date, psyllid control plays a leading role to prevent HLB from spreading. At present, the application of insecticides is the most widely followed option for reducing ACP populations. However, the improper use of insecticide will lead to pest resistance, human poisoning, and environmental pollution [3,4]. Therefore, there is a great need to find environmentally friendly methods to control the citrus psyllid.

During the growth and development of insects, periodic molting, the process of shedding and replacing the rigid insect exoskeleton is required. During this process, the degraded old cuticle is shed and replaced with a newly synthesized cuticle. Insects are covered by this cuticle, which is a composite structure consisting of chitin filaments embedded in a protein matrix. The insect cuticle undertakes various functional roles, including protection, support, movement, and as a shield against environmental stress [5]. Chitin, a polymer of *N*-acetylglucosamine, is an essential component of the cuticle, which plays a critical role in maintaining insect shape and protecting it from external forces [6]. Chitin is mainly found in arthropod exoskeletons, fungal cell walls, and nematode eggshells, but is not found in vertebrates [7]. During insect molting, it is key for the insect to keep a dynamic balance of chitin content, which is regulated by chitin synthase and chitin degradation related enzymes, including chitinase and chitin deacetylase [8]. Based on these crucial roles during insect molting, chitin serves as a selective target for pest control [9].

Chitin formation arises from trehalose and is catalyzed by chitin synthase, a conserved enzyme found in every chitin-synthesizing organism. Chitin synthase belongs to the large family of glycosyltransferases, a ubiquitous group of enzymes that catalyze the transfer of sugar moieties from activated sugar donors to specific acceptors, thereby forming a glycosidic bond [10]. Two chitin synthase (*CHS*) genes have been identified in insects, including *CHS1* and *CHS2*. *CHS1* encodes the isoform of the enzyme for the synthesis of chitin in the cuticle and tracheae, while *CHS2* is produced by midgut epithelial cells for chitin synthesis in the peritrophic membrane (PM) [11]. In recent years, the *CHS* genes have been identified in various insects, including Lepidopteras, Dipterans, Coleopterans, Orthopterans, and Hemipterans [12]. Zhuo et al. identified two chitin synthases genes (*CHS1* and *CHS2*) from *Bombyx mori*, the results suggested that *BmCHS1* is an epidermis-specific gene that is expressed during the molting stage, while *BmCHS2* is midgut specific and highly expressed during the feeding process in the larva [13,14]. In *Tribolium castaneum*, silencing of *CHS1* by RNA interference (RNAi) will disrupt molting and greatly reduce whole-body chitin content, while *CHS2* specific RNAi has no effect on metamorphosis or on total body chitin content [15]. Interestingly, many studies have revealed that *CHS2* is not present in phloem-sucking hemipterans due to the absence of PM [16]. Still, research into the function of *CHS* in *D. citri* has not been reported.

As insects grow in increments, and each stage of growth ends with molting, the inhibition of chitin synthesis is considered as an attractive target site for insect control [10]. Insecticides based on chitin synthesis inhibition are safe to humans as chitin synthesis pathway is absent in vertebrates. Diflubenzuron (DFB) was the first commercial insecticide that acts by inhibiting chitin synthesis in insects [17]. In *Aphis glycines*, when fed with soybean leaves previously dipped in 50 ppm DFB solution, the nymphs suffered significantly higher mortality compared to controls [18]. In *Toxoptera citricida*, fourth-instar nymphs were exposed to 5 and 500 mg/L concentrations of DFB for 48 h and had the highest mortality at the 500 mg/L concentration [19]. Tiwari et al. have revealed that DFB could effectively suppress *D. citri* adult emergence, and fifth-instar nymphs are more susceptible than first-third instar nymphs [4]. However, the exact mode of action of DFB in *D. citri* still remains unclear.

In this study, we identified a cDNA encoding the whole ORF of a chitin synthase (*D. citri CHS (DcCHS*)) from *D. citri* and analyzed the expression patterns of *DcCHS* in different tissues and at different developmental stages. Fluorescence in situ hybridization (FISH) analysis also suggested that the *DcCHS* was evenly distributed in whole integument. The mortality of fifth-instar nymphs significantly increased after exposed to DFB. Furthermore, we demonstrated that dsRNA-mediated gene-specific silencing resulted in a strong reduction in relative expression of *DcCHS* and molting rate of nymphs, as well as an increase in mortality. These results will provide a new target for citrus psyllid control.

## 2. Results

### 2.1. Analysis of the cDNA and Protein Sequences of DcCHS

The cDNA sequence of DcCHS (XP_017303059) contains an ORF of 3180 bp encoding a protein of 1059 amino acid residues with a predicted MW of 129.9 kDa and pI of 5.02 (Figure 1A). Gene structure analysis showed that DcCHS contains 19 exons and 18 introns (Figure 1B). In terms of protein structure, a total of 14 transmembrane helices, two low-complexity regions, and a coiled-coil region were identified (Figure 1C). A BLASTP search of the NCBI databases indicated that the amino acid sequence of DcCHS shared 42.75%, 42.67%, and 42.30% with *Acyrthosiphon pisum*, *Toxoptera citricida*, and *Aphis glycines*, respectively. Based on the amino acid sequences of CHS from different insect species, a phylogenetic tree was generated using MEGA 5.0 to investigate the evolutionary relationship of DcCHS among the selected insect species. The results showed that DcCHS has a close relationship with the CHSs of sap-sucking hemipterans, including *A. pisum*, *T. citricida*, and *A. glycines* (Figure 2).

### 2.2. Spatiotemporal Expression Profiles of DcCHS

The expression profiles of *DcCHS* in different developmental stages and different tissues were investigated by RT-qPCR. The results showed that the *DcCHS* gene was expressed in all tissues, including midgut, integument, leg, wing, and head. It is notable that *DcCHS* had high expression in the integument, while it had low expression in the midgut. The expression level of *DcCHS* in the integument was 44.3 times of that in the midgut, and its expression in the leg was 25.1 times of that in the midgut (Figure 3). The expression of *DcCHS* decreased from first-instar nymph to third-instar nymph, while it increased from third-instar nymph to fifth-instar nymph, and decreased sharply from fifth-instar nymph stage to adult stage (Figure 3). The expression level of *DcCHS* in the fifth-instar nymph was 3.6 times that of the third-instar nymph stage.

### 2.3. Effect of DFB on D. citri Survival and DcCHS Expression Level

A leaf-dip bioassay was used to assay the effect of DFB on *D. citri* survival and *DcCHS* expression level. After being exposed for 24 h, the cumulative mortality of *D. citri* had no significant change between DFB treatment group and control (Triton-100 treatment group). However, the cumulative mortality of DFB sharply increased after exposure to DFB for 48 h (Figure 4B). In addition, *D. citri* in the DFB treatment group showed an abnormal phenotype after molting and the wing of adult exhibited crimp (Figure 4A). The relative expression level of *DcCHS* was significantly upregulated after DFB exposure at 24 h and 48 h (Figure 4C).

### 2.4. Localization of DcCHS Transcript by FISH

In order to confirm localization of DcCHS transcript in *D. citri*, FISH was performed using DcCHS RNA probes. The results showed that DcCHS *transcript* was evenly distributed in the epidermal cells (Figure 5). The fluorescence signal is the strongest in exocuticle epidermal cells of *D. citri*, indicating DcCHS is mainly involved in synthesis of chitin in new cuticle.

### 2.5. RNAi-Based Silencing of DcCHS and Epidermal Structure Analysis

To determine the effect of *DcCHS* on *D. citri* molting, RNAi was performed by ingestion of dsRNA. At 24 h after feeding *D. citri* with double-stranded *DcCHS* (ds*DcCHS*), the expression level of *DcCHS* was significantly downregulated compared with the controls (double-stranded green fluorescent protein (dsGFP) treatment group) (Figure 6A). In the ds*DcCHS* treatment group, many adults did not complete wing development and presented an abnormal wing phenotype after molting (Figure 6B). Importantly, the cumulative mortality in the ds*DcCHS* treatment group was 27% compared to 10% mortality in the group at 48 hpt. The mortality of *D. citri* increased from 24 hpt to 48 hpt in the treatment group, whereas control group had no significant change (Figure 7A). In addition, the cumulative molting of *D. citri* was 53.3% in the treatment group at 48 hpt. However, in the control group, the molting could reach 86.7% (Figure 7B). These results indicated that RNAi effectively inhibited the relative expression level of *DcCHS* and resulted in the abnormal *D. citri* phenotype.

To further analyze the effect of epidermal structure after silencing of *DcCHS*, TEM observation was conducted at 24 h after RNAi treatment. The results showed that the formation of exocuticle was inhibited compared with the control group, while the structure of endocuticle has no significant change (Figure 8). Insect cuticle is primarily composed of epicuticle and procuticle, and procuticle is comprised of the exocuticle and endocuticle. Chitin is found mainly in exocuticle and endocuticle. These results suggested that knockdown of *DcCHS* will inhibit the synthesis of chitin and further disrupt the structure of *D. citri* cuticle.

## 3. Discussion

Chitin is a major component of the exoskeleton and the peritrophic matrix of insects. The synthesis of chitin is catalyzed by many enzymes. Among them, chitin synthases (CHSs) play important roles during insect development and metamorphosis [8]. To date, *CHS* genes have been identified from many insect species, including those in the Lepidoptera, Hemiptera, Hymenoptera, Diptera, and Coleoptera. The number of genes encoding *CHS* in different fungal species can range from 1 to 20, while in insect genomes characterized so far individual species contain only two *CHS* genes (*CHS-1* and *CHS-2*) [20]. *CHS-A* is responsible for chitin synthesis in the cuticle and cuticular lining of the foregut, hindgut, and trachea, whereas *CHS-B* is dedicated to chitin synthesis in the PM [21]. In this study, only one *CHS* gene was identified from the genome of *D. citri*, which was named *DcCHS* (Figure 1). In a previous report, many hemipterous insects only contain a single *CHS* gene, including *A. glycines*, *Sogatella furcifera*, and *A. pisum* [18]. We considered that Hemiptera insects lack the PM structure. Terry et al. revealed that the insect PM was lost, leading to the compartmentalization of the digestive process and ultimately increased digestion of polymers during the course of evolution [22]. The insect CHS proteins usually contain 15 transmembrane helices that flank the catalytic domain located on the cytoplasmic side of the plasma membrane [10]. We found that DcCHS contains 14 transmembrane helices (Figure 1). Depending on the number of predicted transmembrane helices, the N-terminus faces either the extracellular space or the cytoplasm. However, the C-terminus region is predicted to face the extracellular space and may be involved in protein–protein interaction or oligomerization [5,23]. In addition, DcCHS has two conserved domains with chitin synthase signature motifs, i.e., EDR and QRRRW in the catalytic domain, which is essential for the catalytic mechanism [24]. The phylogenetic tree analysis indicated that *CHS* from *D. citri* has a close relationship with *A. pisum* and belongs to the CHS1 group. Interestingly, many studies have proved that the *CHS1* gene contains alternative splicing [25,26]. In the present study, the genome analysis of *D. citri* revealed no splice variants exist. Bansal et al. also found that splice variants of *CHS1* do not exist in the *A. glycines* genome [18]. However, the specific reasons for this apparent loss need to be researched in depth.

The relative expression level of *DcCHS* was determined in different tissues and different developmental stages. Results suggested that *DcCHS* had a relatively higher expression in the integument, followed by the leg, but it had low expression in the midgut (Figure 3). Our results are in agreement with the roles of CHS in chitin production in the insect exoskeleton [27]. In insects, chitin functions as a scaffold material, supporting the cuticles of the epidermis [5]. In *Sogatella furcifera*, *CHS1* was also predominantly expressed in the integument [28]. Therefore, we speculated that *DcCHS* may play a critical role in the process of cuticle formation. Additionally, the phenomenon of low expression of *CHS* in the midgut was also observed in *Plutella xylostella* and *Nilaparvata lugens* [6,29]. At different developmental stages, *DcCHS* had a higher expression in the fifth-instar nymph stage (Figure 3). The fifth-instar nymph stage is a critical period that involves progressing from nymph stage into adult stage [30]. During molting period, the synthesis of chitin is required to maintain rigid structure of new cuticle in *D. citri*.

Chitin is synthesized by insects and fungi but not by vertebrates, so chitin synthase presents a novel target for pest control. In recent years, the exploration of inhibitors of chitin synthesis has received extensive attention [17]. DFB is an insect growth regulator acting on chitin synthesis, which was discovered in the 1970s [31]. It was used as a potential insecticide for pest control in forestry and agriculture [32]. In a previous research, Tiwari et al. revealed that DFB could effectively suppress *D. citri* adult emergence [4]. However, the specific molecular mechanisms are unclear. In this study, we found that DFB can increase the expression of *DcCHS* and lead to higher mortality of *D. citri* (Figure 4). In many insect species, DFB has been found to elevate *CHS* expression level. Zhang et al. revealed a significant increase of *CHS1* mRNA level in *Anopheles quadrimaculatus* larvae exposed to DFB at 100 and 500 μg/L [33]. In *T. citricida*, the mRNA expression levels of *TCiCHS* were significantly increased upon the exposure of nymphs to both low and high DFB concentrations [19]. At 48 h post DFB treatment (hpt), the cumulative mortality in the treatment group was significantly greater than the control group, whereas it showed no obvious change at 24 hpt. Therefore, we speculated that DFB can reduce the chitin content by inhibition of chitin synthase activity. The increasing *DcCHS* expression levels may also indicate the existence of a feedback regulatory mechanism that compensates for the low enzyme content.

RNA interference (RNAi) has already proven its usefulness in functional genomics research on insects, but it also has considerable potential for the control of pest insects [34]. When monitoring RNAi responses using different delivery methods, variable efficiency is very common in different insect species, e.g., the physiological pH of hemolymph significantly affected the efficiency of RNAi in *Locusta migratoria* [35]. In this study, we performed RNAi experiments through the feeding of dsRNA. The results showed that *DcCHS* was silenced effectively (Figure 5). In recent years, RNAi combined with RT-qPCR has been widely used to research the functions of target genes. Yu et al. revealed that silencing of *transformer-2* gene influenced female reproduction and offspring sex in *D. citri* [36]. Kishk et al. used RNAi to silence genes implicated in pesticide resistance in order to increase susceptibility, and the results suggested that the treatments with dsRNA caused concentration-dependent nymph mortality [37]. In addition, we found that *D. citri* showed increased cumulative mortality and abnormal phenotypes after silencing of *DcCHS* (Figure 6). In this regard, many reports on RNAi mediated knockdown of insect *CHS* genes resulting in lethal phenotypes are encouraging [15,38,39]. Chitin is the main component of exocuticle and endocuticle [40]. We found that the formation of exocuticle was inhibited after knockdown of *DcCHS*, while the structure of endocuticle had no significant change, indicating the decrease of chitin content will influence the structure of cuticle. These results further indicated that *DcCHS* could be used as a new target for control of *D. citri*.

## 4. Materials and Methods

### 4.1. Diaphorina citri Rearing and Collection

The ACP was reared at the National Navel Orange Engineering Research Center (NORC), Gannan Normal University, Ganzhou, China. *D. citri* was reared in mesh cages (50 × 50 × 50 cm) on *Murraya exotica* with 27 ± 1 °C, 70 ± 5% relative humidity, and a 14:10 (light: dark) photocycle.

*D. citri* nymphs were classified to different stages according to their morphological features and collected using an aspirator. *D. citri* adults were divided into three groups, and then each group was dissected to obtain the midgut, head, leg, wing, and integument. The collected tissues were cleaned with DEPC water and stored at −80 °C until use. Each experiment was conducted in three biological replicates.

### 4.2. RNA Isolation and cDNA Synthesis

To obtain the cDNA template for spatiotemporal expression analysis of *DcCHS*, total RNA was isolated from different tissues of adult insects (head, midgut, leg, wing, and integument) and at different developmental stages of the nymph (first-instar, second-instar, third-instar, fourth-instar, and fifth-instar nymphs) using a RNA extraction kit (TaKaRa Biotechnology Co. Ltd., Dalian, China). The integrity of total RNA was confirmed using standard agarose gel electrophoresis with ethidium bromide staining. RNA quantification was performed using a NanoDrop 2000 spectrophotometer (Thermo Fisher Scientific, New York, NY, USA). The purity of all RNA samples was assessed at an absorbance ratio of A_260/230_ and A_260/280_. Total RNA was reverse-transcribed in a 20 μL reaction system using a cDNA synthesis master mix kit according to the manufacturer’s instructions (Simgen, Hangzhou, Zhejiang, China). In brief, 2.0 μL of 5 × gRNA buffer and 1 μg of total RNA were mixed, then RNase-free water was added to reach 10 μL, which was then incubated at 42 °C for 3 min. Afterward, 4 μL of 5 × RT buffer, 2 μL of RT enzyme mix, and 4 μL RT primer mix were added to the tube. The mixture was incubated at 42 °C for 15 min and then incubated at 95 °C for 3 min. The cDNA was stored at −20 °C for later use.

### 4.3. Identification of DcCHS and Bioinformatics Analysis

To identify chitin synthase genes in *D. citri*, sequences of *Nilaparvata lugens CHS* (*NlCHS*: AEL88648.1) and *A. pisum CHS* (*ApCHS*: XP_003247517.1) were used as query in a TBLASTN search of genome database of *D. citri*. The cDNA and deduced amino acid sequence of DcCHS were analyzed by using DNASTAR. The open reading frame (ORF) was identified according to the ORF finder tool (https://www.ncbi.nlm.nih.gov/orffinder/) at the National Center for Biotechnology Information (NCBI). The molecular weight (MW) and isogenic point (pI) of DcCHS were calculated using ExPASy (http://web.expasy.org/compute_pi). The signal peptide of DcCHS was predicted using SignalP 4.1 Server (http://www.cbs.dtu.dk/services/SignalP). The functional domain was predicted by using SMART software (http://smart.embl-heidelberg.de/). The membrane-spanning domain was predicted by TMHMM Server v. 2.0. The phylogenetic tree was constructed with MEGA 7.0 software using the neighbor-joining method with 1000-fold bootstrap resampling. Protein sequences from different species were downloaded from GenBank (http://www.ncbi.nlm.nih.gov/) and used in phylogenetic analysis (Appendix A).

### 4.4. RT-qPCR Analysis of DcCHS Expression Levels

RT-qPCR was conducted to confirm the relative expression levels of *DcCHS* in various tissues and developmental phases. The primers were designed using Primer Premier 5.0 software (Premier Biosoft, www.premierbiosoft.com) (Table 1). The 20 μL of reaction mixture for the RT-qPCR contained 10 μL of SYBR II, 8 μL of ddH_2_O, 0.5 μL of forward primer, 0.5 μL of reverse primer, and 1.0 μL of cDNA template. The thermal cycling profile consisted of an initial denaturation at 95 °C for 60 s, and 40 cycles of 95 °C for 10 s, 60 °C for 10 s, and 72 °C for 10 s. The reactions were performed with a LightCycle^®^96 PCR Detection System (Roche, Basel, Switzerland). Relative expression levels were calculated by using the 2^−ΔΔ*C*t^ method. There were three biological replicates and three technique replicates for each sample. The reference gene chosen for analysis of *DcCHS* in different tissues and different developmental phases was *glyceraldehyde-3-phosphate dehydrogenase* (*GAPDH*). This gene was cited as a reference in previous studies involving expression analyses in *D. citri* [41]. All ANOVAs were followed by Fisher’s protected least significant difference (LSD) tests.

### 4.5. Leaf-Dip Bioassay

To determine whether DFB can affect the expression of *DcCHS* and increase the mortality of *D. citri*, we used a leaf-dip bioassay according to a previous protocol with some modifications [32]. In brief, about 0.05 g of DFB (LKT Laboratories Inc., Saint Paul, MN, USA) was added into 10 mL acetone and dissolved completely as a stock solution (5000 mg/L). The stock solution was diluted into a low concentration (300 mg/L) using ddH_2_O, and 50 μL of Triton-100 (0.1%) was added. Detached tender leaves of *Murraya koenigii* were dipped into the DFB working solution for 5 min and air-dried for 1 h, after which 40 fifth-instar *D. citri* nymphs were transferred to each leaf. Leaves treated with ddH_2_O containing 0.1% Triton-100 were used as a control. The number of deaths and molts were counted at 24 h and 48 h after DFB exposure. In addition, the surviving *D. citri* were kept at −80 °C for RNA isolation. The relative expression levels of *DcCHS* were analyzed by RT-qPCR as described above. All experiments were performed using three biological replicates.

### 4.6. dsRNA Synthesis and DcCHS RNAi Analysis

The primers for *DcCHS* were designed to synthesize dsRNA using the T7 RioMAX Express RNAi System (Promega, San Luis Obispo, CA, USA) based on the manufacturer’s instructions. GFP dsRNA was used as a control. The dsRNA delivery was performed by using an artificial diet according to a previous method [42]. In brief, 30 newly emerged fifth-instar *D. citri* nymphs were used in dsRNA treatment. All nymphs were allowed to feed on an aliquot of the artificial diet (150 μL) placed between two layers of stretched Parafilm. The artificial diet consisting of 20% (*w*:*v*) sucrose was mixed with ds*DcCHS* at a final concentration of 150 ng/μL. After 24 h, all live insects were collected to isolate total RNA and synthesize cDNA. The effect of ds*DcCHS* on gene expression was evaluated by RT-qPCR. A total of three biological replicates were used for each experiment.

### 4.7. Fluorescence In Situ Hybridization (FISH) Analysis

FISH was performed to confirm the specific distribution of *DcCHS* transcript in *D. citri*. The RNA probes of *DcCHS* (5′-Cy3-CGUAAGUCCUUCAAAUCGCUCGUAAUUCGACUCUG-3′) were synthesized by Rochen Biotechnology (Rochen, Shanghai, China). FISH was conducted according to previous protocol with some modification [43]. In brief, the dissected integument samples were fixed in 4% paraformaldehyde for about 8 h at 4 °C and then washed using PBST (PBS+Triton X-100) for three times (5 min each), treated with proteinase K (50 μg/mL in PBST). The fixed tissues were prehybridized in hybridization buffer at 37 °C for 1 h and then hybridized in the same hybridization buffer containing 10 μg/mL RNA probes overnight at 37 °C. After hybridization, the samples were washed using 2 × SSC at 37 °C for 10 min, 1 × SSC at 37 °C for 5 min, and 0.5 × SSC at 37 °C for 10 min to remove remaining hybridization buffer, and then moved to a fresh microscope slide containing 30 μL of new hybridization buffer supplemented with DAPI and kept with a liquid blocker. At last, the samples were visualized under an Olympus fluorescence microscope equipped with Cy3 filter set.

### 4.8. Transmission Election Microscopy (TEM) Analysis

To investigate the transformation of epidermal structure after silencing of *DcCHS*, TEM was used following a previous report [21]. Abdominal integument of a hatched adult was dissected and fixed in 2.5% glutaraldehyde of PBS at 4 °C overnight and then fixed in 1% osmic acid at 4 °C for 3 h. After fixation, all samples were washed three times using 0.1 M PBS. The fixed integuments were dehydrated through incubation in a graded series of ethanol washes (50%, 70%%, 80%, 85%, 90%, 95%, and 100%, *v*/*v*) for 15 min each and then further dehydrated twice using 100% ethanol for 10 min each. The dehydrated samples were consecutively soaked by penetrant 1 (2:1 mixture of acetone and epoxy resin), penetrant 2 (1:1 mixture of acetone and epoxy resin), and penetrant 3 (epoxy resin) at 37 °C for 12 h. Finally, the samples were embedded at 60 °C for 48 h. The ultrathin (100 nm) sections were cut with a Leica microtome (Leica, Wetzlar, Germany). The sections were stained with 3% uranyl acetate and alkaline lead citrate and observed using TEM with a mode (Tecnai G2 20 S-TWIN, EDX: GENESIS 2000) at an accelerating voltage of 200 kV.

## 5. Conclusions

In conclusion, the cDNA sequence of *DcCHS* was identified from the genome database of *D. citri*. Spatiotemporal expression analysis suggested that *DcCHS* was highly expressed in the integument and fifth-instar nymph stage. In addition, *DcCHS* was upregulated and the cumulative mortality of *D. citri* increased after exposure to DFB, an inhibitor of chitin synthesis. Furthermore, RNAi-based gene silencing inhibited the expression of *DcCHS* and influenced the structure of cuticle, resulting in malformed phenotypes. Taken together, these results indicated *DcCHS* as a new target for control of *D. citri*.

## Figures and Tables

**Figure 1 ijms-20-03734-f001:**
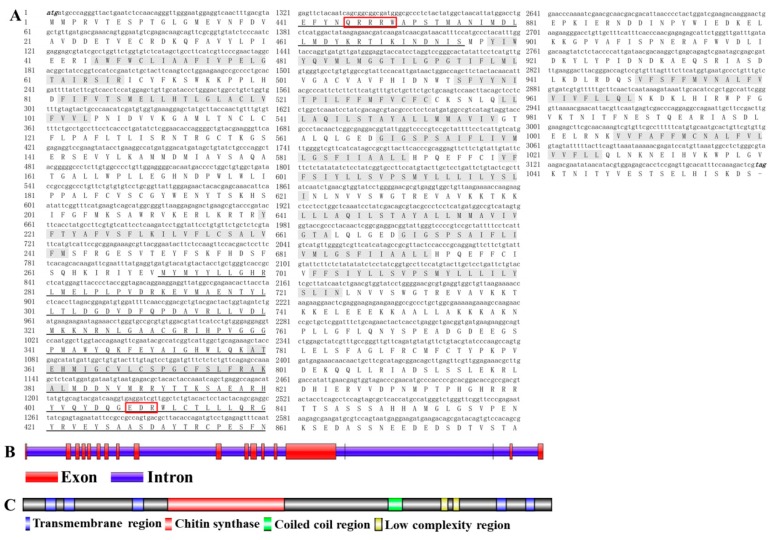
Bioinformatics analysis of *Diaphorina citri* chitin synthase (*DcCHS*). (**A**) Complete nucleotide and deduced amino acid sequence of the *DcCHS* cDNA. Numbers on the left side represent nucleotide and amino acid positions. The initiation codon (ATG) and termination codon (TAG) are indicated in black italics. The Chitin_synth_2 domain is indicated in a single line. (**B**) Gene structure analysis of *DcCHS* by using Splign online software. (**C**) Domain analysis of DcCHS by Illustrator for Biological Sequences (IBS) software.

**Figure 2 ijms-20-03734-f002:**
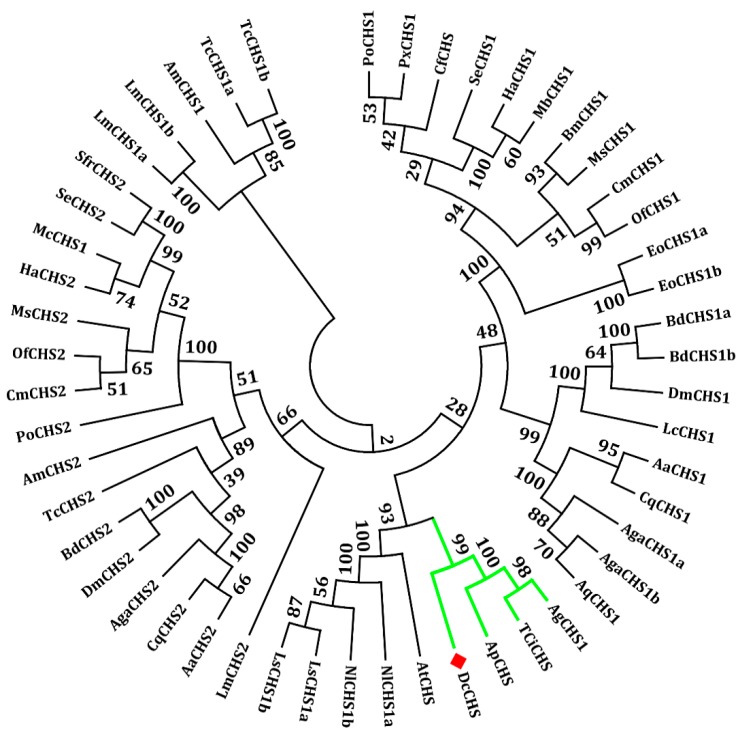
Phylogenetic relationships of DcCHS in different insect species using the neighbor-joining method with a bootstrap value of 1000. Chitin synthases were from *Acyrthosiphon pisum* (Ap), *Aphis glycines* (Ag), *Laodelphax striatellus* (Ls), *Nilaparvata lugens* (Nl), *Bombyx mori* (Bm), *Choristoneura fumiferana* (Cf), *Cnaphalocrocis medinalis* (Cm), *Toxoptera citricida* (TCi), *Aedes aegypti* (Aa), *Culex quinquefasciatus* (Cq), *Drosophila melanogaster* (Dm), *Ectropis obliqua* (Eo), *Helicoverpa armigera* (Ha), *Mamestra brassicae* (Mb), *Apis mellifera* (Am), *Phthorimaea operculella* (Po), *Anopheles gambiae* (Aga), *Anopheles quadrimaculatus* (Aq), *Lucilia cuprina* (Lc), *Locusta migratoria manilensis* (Lm), *Manduca sexta* (Ms), *Plutella xylostella* (Px), *Spodoptera exigua* (Se), *Tribolium castaneum* (Tc), *Ostrinia furnacalis* (Of), *Spodoptera frugiperda* (Sf), *Anasa tristis* (At), and *Bactrocera dorsalis* (Bd).

**Figure 3 ijms-20-03734-f003:**
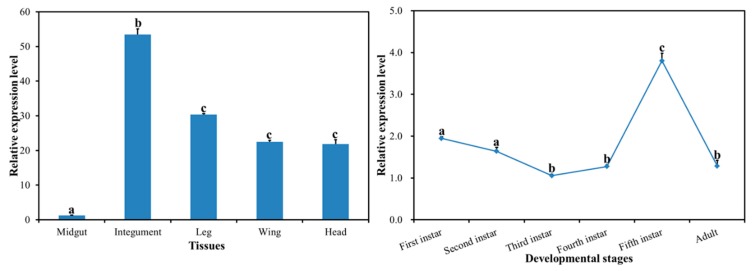
Expression profiles of *DcCHS* at different developmental stages and in different tissues of the adult of *D. citri*. Relative mRNA levels of *DcCHS* as examined using RT-qPCR. Data were normalized using glyceraldehyde-3-phosphate dehydrogenase (*GAPDH*) and are represented as the means ± standard errors of the means from three independent experiments. Relative expression levels were calculated using the 2^−∆∆Ct^ method. Statistical analysis was performed using SPSS software. The significant differences are indicated by a different letter, for example, a, b, and c (*P* < 0.05).

**Figure 4 ijms-20-03734-f004:**
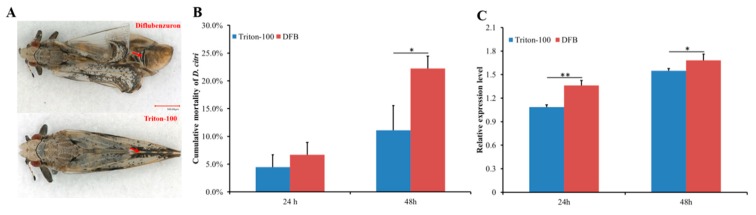
Effect of diflubenzuron (DFB) on *Diaphorina citri* morphology, mortality rate, and DcCHS expression levels. (**A**) Phenotypes of *D. citri* at 24 h of DFB treatment and Triton-100. (**B**) Cumulative mortality of *D. citri* following feeding with DFB and Triton-100 (Control). (**C**) Relative expression of *DcCHS* in *D. citri* feeding with DFB and Triton-100 (Control). The mean expression level represented for three biological replicates. Relative expression levels were calculated using the 2^−∆∆*C*t^ method. Statistical analysis was performed using SPSS software. The significant differences are indicated by * (*P* < 0.05) or ** (*P* < 0.01).

**Figure 5 ijms-20-03734-f005:**
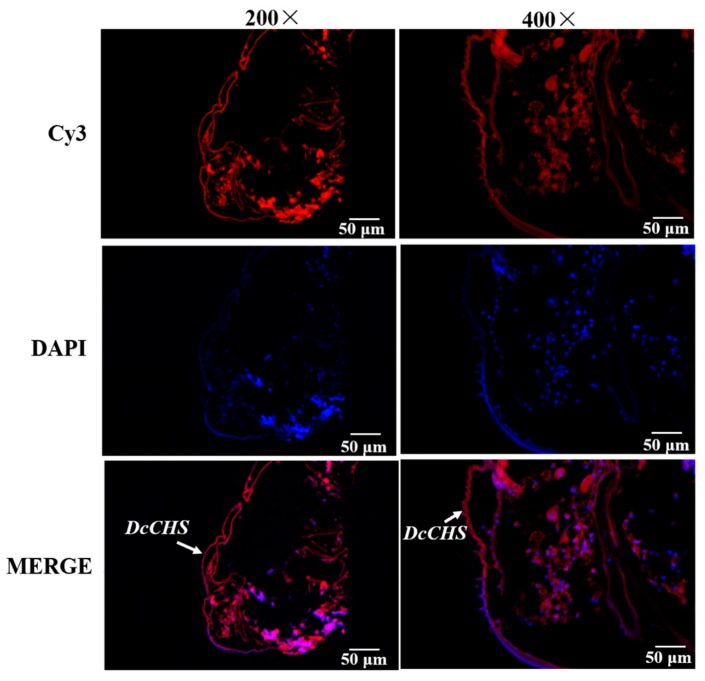
Fluorescence in situ hybridization (FISH) analysis of *DcCHS* in psyllid integument using *DcCHS* probe. The red color indicated the distribution of *DcCHS* transcript. The blue color indicated the distribution of cell nucleus in *D. citri*.

**Figure 6 ijms-20-03734-f006:**
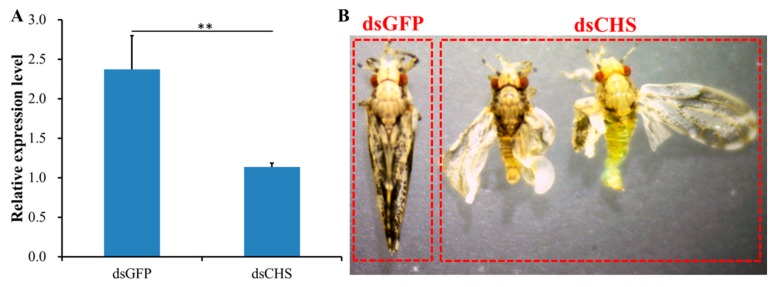
Effects on *Diaphorina citri chitin synthase* (*DcCHS*) after RNA interference (RNAi) of *DcCHS*. (**A**) Relative expression levels of *DcCHS* when *D. citri* was treated with double-stranded CHS (ds*DcCHS*) and double-stranded green fluorescent protein (ds*GFP*). The mean expression level represented for three biological replicates. Relative expression levels were calculated using the 2^−∆∆*C*t^ method. Statistical analysis was performed using SPSS software. The significant differences are indicated by ** (*P* < 0.01). (**B**) Representative phenotypes of *D. citri* at 24 h post RNAi treatment.

**Figure 7 ijms-20-03734-f007:**
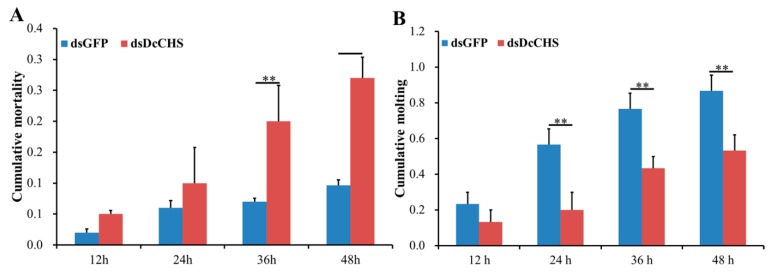
Effects on survival of *D. citri* after RNA interference (RNAi) of *DcCHS*. (**A**) Cumulative mortality of *D. citri* in the treatment (ds*DcCHS*) and control (ds*GFP*) groups at 12, 24, 36, and 48 h post RNAi treatment. (**B**) Cumulative molting of *D. citri* in the treatment (ds*DcCHS*) and control (ds*GFP*) groups at 12, 24, 36, and 48 h post RNAi treatment. The significant differences are indicated by ** (*P* < 0.01).

**Figure 8 ijms-20-03734-f008:**
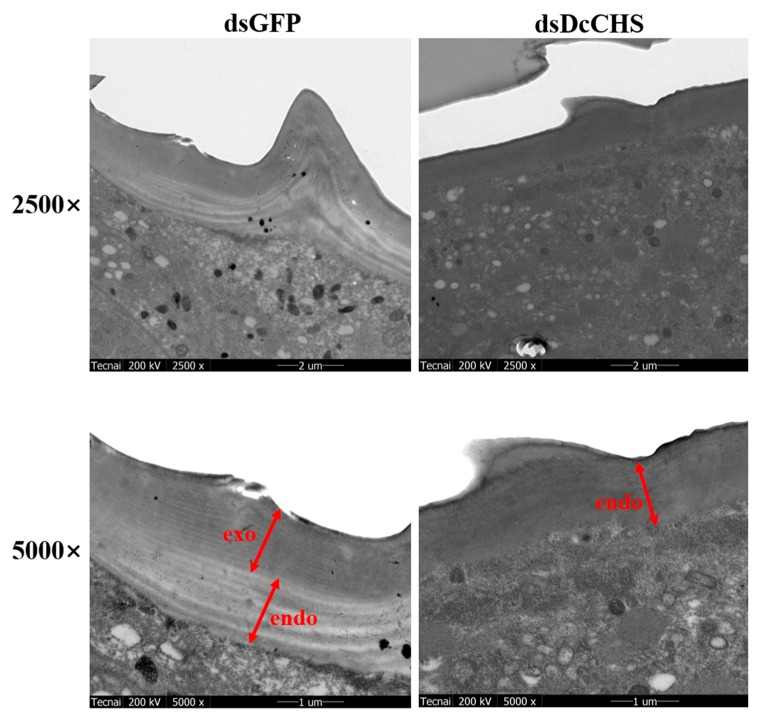
TEM observation of the integument of adult *D. citri* treated with ds*DcCHS*. *D. citri* treated with ds*GFP* as a control. Images were acquired using a Tecnai TEM. Three biological replicates were performed for each condition. (Endo: endocuticle; Exo: exocuticle).

**Table 1 ijms-20-03734-t001:** Primers used in this study.

Primers	Sequences	Purpose
DcCHS-RT-F	TCAGCATGGCGGGTTAAG	RT-qPCR
DcCHS-RT-R	CTCCGCGGAATGACATGAATA	
GAPDH-F	CATGGCAAGTTCAACGGTGA	
GAPDH-R	CGATGCCTTCTCAATGGTGG	
ds-DcCHS-F	TAATACGACTCACTATAGGGAGACAGGAAGGAGGTTATG	dsRNA synthesis
ds-DcCHS-R	TAATACGACTCACTATAGGGCATCTGGTGTAAGCGTCA	
ds-GFP-F	GGATCCTAATACGACTCACTATAGGCAGTGCTTCAGCCGCTACCC	
ds-GFP-R	GGATCCTAATACGACTCACTATAGGACTCCAGCAGGACCATGTGAT	

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
