# Peer review of "Silencing of the Chitin Synthase Gene Is Lethal to the Asian Citrus Psyllid, Diaphorina citri"

_ijms, 2019, doi:10.3390/ijms20153734_

Round 1

Reviewer 1 Report

1) Line 57: CHS1 and CHS2 are two distinct genes. The description "The CHS gene can be divided into CHS1 and CHS2" may give readers the impression that CHS1 and CHS2 are alternatively spliced variants of the same gene. To avoid confusion, it would be better to state that two CHS genes, CHS1 and CHS2, have been identified in insects.

2) Line 59: "CHS gene" should be "CHS genes".

3) Line 75: The genus name (Toxoptera) should be given here as it is the first mention of the species in the text.

4) Line 96: "Aphis Glycines" should be "Aphis glycines" (no capitalization of species name).

5) Line 103 (Figure 1 caption): "DcCHS gene" should be "DcCHS cDNA" for the sequence shown in part A.

6) Lines 159-162: The localization of DcCHS transcript to the integument is not evident in Fig 5 without a visible light image for comparison as it is impossible for the reader to know what they are looking at. Furthermore, since the technique relies on detection of the DcCHS transcript, the signal should only be seen in the epidermal cells and not in the cuticle, and certainly not in the exocuticle (as indicated on line 161) which is the more outer layer of the procuticle. (Since the cuticle and underling epidermal cells can collectively be referred to as the "integument", it is not improper to use that description here.)

7) Lines 169-170: "After feeding dsDcCHS at 24 h, the expression level of DcCHS was significantly up-regulated compared with the controls"; should this instead read "down-regulated"? 

8) More description needs to be given for the TEM images in Fig 8. For the "Blank" images it appears that the epidermis is at the top of the images, therefore, what is the homogeneous grey structure at the bottom? If the sections were prepared from adults after molting I would not expect to see any kind of a structure outside of the cuticle.  For the 5000X magnification of the dsGFP sample it appears that the labeling of the exo- and endocuticle are reversed; the endocuticle should be the inner layer adjacent to the epidermis while the exocuticle is the outer layer.

9) Lines 204-205: This would be better if written as "To date, CHS genes have been identified from many insect species ..." as many insects have multiple CHS genes.

10) Line 280: Is the correct genus for the host plant Murray or Murraya?

11) Lines 310-311: The sentence concerning the use of Signal P seems to be incomplete.

12) Line 328: "glceraldehyde" should be "glyceraldehyde".

13) Line 329: Citations for the use of GADPH as the reference gene should be given here.

14) Line 340: The heading for section 4.5 is the same as that for section 4.4 instead of reflecting the analyses with DFB.

15) Line 368: The previous protocol followed for FISH analysis should be cited here.

16) Table S1: If the DcCHS sequence shown in Figure 1 is the same as XP_017303059 then that accession number should also be given in Table S1 (instead of left blank).

Author Response

1.       Line 57: CHS1 and CHS2 are two distinct genes. The description “The CHS gene can be divided into CHS1 and CHS2” may give readers the impression that CHS1 and CHS2 are alternatively spliced variants of the same gene. To avoid confusion, it would be better to state that two CHS genes, CHS1 and CHS2, have been identified in insects.

Reply: Thanks for reviewer’s valuable and thoughtful suggestions. We have revised the related descriptions in previous manuscript, seeing Line 58 to Line 59

2.       Line 59: “CHS gene” should be “CHS genes”.

Reply: Thanks for reviewer’s valuable comments. We have revised “CHS gene” into “CHS genes” in previous manuscript, seeing Line 62.

3.       Line 75: The genus name (Toxoptera) should be given here as it is the first mention of the species in the text.

Reply: Thanks for reviewer’s thoughtful comments. We have added the genus name (Toxoptera) in previous manuscript, seeing Line 77.

4.       Line 96: "Aphis Glycines" should be "Aphis glycines" (no capitalization of species name).

Reply: Thanks for reviewer’s thoughtful comments. We have revised “Aphis Glycines” into “Aphis glycines” in previous manuscript, seeing Line 99 and Line 103.

5.       Line 103 (Figure 1 caption): "DcCHS gene" should be "DcCHS cDNA" for the sequence shown in part A.

Reply: Thanks for reviewer’s valuable comments. We have revised “DcCHS gene” into “DcCHS cDNA” in previous manuscript, seeing Line 106.

6.       Lines 159-162: The localization of DcCHS transcript to the integument is not evident in Fig 5 without a visible light image for comparison as it is impossible for the reader to know what they are looking at. Furthermore, since the technique relies on detection of the DcCHS transcript, the signal should only be seen in the epidermal cells and not in the cuticle, and certainly not in the exocuticle (as indicated on line 161) which is the more outer layer of the procuticle. (Since the cuticle and underling epidermal cells can collectively be referred to as the "integument", it is not improper to use that description here.)

Reply: Thanks for reviewer’s valuable and thoughtful comments. We have revised the improper descriptions in previous manuscript, seeing Line 165 to Line 167. In this study, we performed FISH experiment using DcCHS RNA probes. Because the individual size of citrus psyllid is comparatively tiny, it is very different to obtain undamaged cuticle section. We performed multiple replication to clear light image, and then conducted FISH experiment. We have provided a visible light image in the attachment.

7.       Lines 169-170: "After feeding dsDcCH at 24 h, the expression level of DcCHS was significantly up-regulated compared with the controls"; should this instead read "down-regulated"?

Reply: Thanks for reviewer’s valuable comments. We have revised the incorrect descriptions in previous manuscript, seeing Line 176.

8.       More description needs to be given for the TEM images in Fig 8. For the "Blank" images it appears that the epidermis is at the top of the images, therefore, what is the homogeneous grey structure at the bottom? If the sections were prepared from adults after molting I would not expect to see any kind of a structure outside of the cuticle.  For the 5000X magnification of the dsGFP sample it appears that the labeling of the exo- and endocuticle are reversed; the endocuticle should be the inner layer adjacent to the epidermis while the exocuticle is the outer layer.

Reply: Thanks for reviewer’s thoughtful comments. According to the second reviewer’s suggestions. We have removed the blank in the Figure 8, seeing Line 208 to Line 209. In addition, we are sorry for incorrect descriptions in previous manuscript. After silencing of DcCHS, the formation of exocuticle was inhibited compared with the control group, while the structure of endocuticle has no significant change. We have revised the Figure 8 and relevant descriptions, seeing Line 187 and Line 189.

9.       Lines 204-205: This would be better if written as "To date, CHS genes have been identified from many insect species..." as many insects have multiple CHS genes.

Reply: Thanks for reviewer’s valuable comments. We have revised the related descriptions in previous manuscript, seeing Line 217 to Line 218.

10.   Line 280: Is the correct genus for the host plant Murray or Murraya?

Reply: We are sorry for incorrect descriptions in previous manuscript. We have revised the related descriptions, seeing Line 299.

11.   Lines 310-311: The sentence concerning the use of Signal P seems to be incomplete.

Reply: We are sorry for incorrect descriptions in previous manuscript. We have added the detailed descriptions, seeing Line 329 to Line 330.

12.   Line 328: "glceraldehyde" should be "glyceraldehyde".

Reply: Thanks for reviewer’s thoughtful comments. We have revised “glceraldehyde” into “glyceraldehyde”, seeing Line 348.

13.   Line 329: Citations for the use of GADPH as the reference gene should be given here.

Reply: Thanks for reviewer’s thoughtful suggestions. We have added the related references in previous manuscript, seeing Line 350.

14.   Line 340: The heading for section 4.5 is the same as that for section 4.4 instead of reflecting the analyses with DFB.

Reply: Thanks for reviewer’s thoughtful comments. We have revised the related descriptions, seeing Line 360.

15.   Line 368: The previous protocol followed for FISH analysis should be cited here.

Reply: Thanks for reviewer’s valuable comments. We have added the related references in previous manuscript, seeing Line 377.

16.   Table S1: If the DcCHS sequence shown in Figure 1 is the same as XP_017303059 then that accession number should also be given in Table S1 (instead of left blank).

Reply: Thanks for reviewer’s valuable suggestions. We have added the accession No of DcCHS in Table S1. 

Reviewer 2 Report

In this paper, the researchers identified a chitin synthase gene (CHS) in the Asian citrus psyllid (ACP), and used their newly discovered gene to explore two control options for the pest. The first control option was Diflubenzuron (DFB), which had already been demonstrated to kill ACP; however, the authors demonstrate that, like in other species, DFB kills by impacting chitin production, and that CHS expression increases in response. The authors also look at CHS RNAi as a potential control method, and find that, like DFB, CHS RNAi disrupts chitin production, causing mortality. These findings will be of value to the ACP research community, as they continue to work to bring ACP under control, and limit the damage it is causing to citrus groves around the world. The authors’ research and conclusions appear to be sound.

However, there are several minor issues that detract from the impactfulness of this paper. Overall, the language was readable, but I identified several minor errors that were distracting. I will not enumerate all of these errors here, but I encourage the authors to get a native English speaker to give their paper a read-through before making their final submission.

I also identified several issues with how the data were presented and discussed. Most of these errors, while egregious, should be simple to fix. I will discuss them below in the order they were encountered in the paper:

Abstract: Line 23: the authors discuss fifth-instar nymphs exposed to DFB having “the highest mortality.” However, the sentence has no context for what this is compared to. Having read the paper, I believe they mean that mortality was highest in fifth-instar nymphs relative to the other tested life stages. This should be clarified.

Introduction: Line 83. “Fifth-instar nymphs exposed to DFB further increased mortality.” This sentence provides no affirmation that the authors are repeating already published work. Moreover, the word ‘increased’ has no context, and its use as a verb with ‘nymphs’ as a subject doesn’t really make sense. It isn’t clear to me if the authors mean to imply that they got higher mortality rates that prior work, or that their new research found a better target age. This sentence should be rewritten to clarify these issues.

Results:

Line 97: the authors report the use of MEGA 5.0, which is several versions old. If possible, the authors should consider updating their tree with the latest version of MEGA.

Figure 1A: From an aesthetic point of view, the sequence presentation would look better if presented in more even columns; while preferable, this is not necessary. However, the authors claim to be providing the complete nucleotide sequence, yet all the nucleotide sequence they provide in Figure 1 is translated. If there is no 5’ or 3’ UTR, the authors should state this. If there is UTR, it should be included in Figure 1A. Finally, the authors state in the figure legend that the stop codon of their sequence is TAA, but that actually shown in the image is TAG. This discrepancy should be corrected.

Figure 2: The text in the figure is difficult to read. At the very least, the text should be enlarged. Alternatively, the figure can be made bigger; this might work better if the figure was in box rather than circle format.

Figure 3: The Y-axis on each graph is described as "Relative expression level" but this requires a comparison to some baseline. To what are the expression levels "relative"? Do the authors mean "normalized" instead? If the latter, then the Y-axis label should be changed to reflect this reality. For this figure, the authors also provide statistical comparisons to highlight significant differences. However, in each graph, multiple categories are shown, yet, the authors do not clarify, either in the figure legend or in the text, what each category is compared to. This should be explained, since these are cases where the baseline for comparison is intuitive. Alternatively, the authors could use a letter system to indicate categories that are similar.

Line 144 and following: The authors describe their DFB leaf-dip assay in some detail here and in the Materials and Methods, but what is not clear at any point is whether they exposed their test insects to the DFB for a short period or for the entire 24-48 hrs. This should be made clear in the Materials and Methods, and the language in this paragraph and in the rest of the paper should be altered accordingly. E.g. on line 144, instead of “At 24 h”, the authors could say, “Twenty-four hours after initial exposure” or “After being exposed for 24 h” depending on which method they employed. Additionally, some language implies that insects were re-exposed for the 48-hr test. This should also be clarified. E.g. on line 146, instead of “sharply increased after exposure [to] DFB at 48 h” the authors should say “after a second exposure at 48 h” or “after exposure to DFB for 48 h”.

Line 151: I recommend replacing the word “phenotype” with “morphology”

Line 169-170: the authors describe DcCHS expression as “significantly up-regulated” following RNAi. However, this is the opposite of what would be expected, and is also contradictory to the data presented in Figure 6A. Most likely, this is a misstatement and should be corrected to say “significantly reduced”

Line 175 and Figure 7: the authors mention “cumulative molting rate”, but they are not really showing a rate so much as total percent of adults that have molted. The authors would be fine removing the word ‘rate’ from the text and the figure.

Figure 7: The legend does not discuss the significance values indicated, or how they were calculate. Moreover, the authors do not seem to be making appropriate comparisons. In graphs like this, researchers will typically show how significantly different the Treatment is relative to the Control, not how different one time point is from another. Since the results are cumulative, one would expect that, after sufficient time, enough individuals are affected to create a significant time point, after which, every other time point would be significant. Therefore, making significance comparisons between later time points and the start, as the authors seem to be doing, is meaningless. What is more important is if and when the numbers in the Treatment significantly diverge from the numbers in the Control. The authors should redo their statistical tests to be more in line with this expectation.

Figure 8: The authors do not explain what the blank is. Moreover, it looks very different even from their control (almost like it is a lower magnification). The authors need to explain what this image is meant to show and why it looks different from control. If the image serves no effective purpose, then it should be removed – the control (dsGFP) should be sufficient.

Discussion:

Line 233-234: this sentence seems internally redundant, and its intended meaning (and role in progressing the authors’ arguments) is not immediately clear. This sentence should be reworded to be clearer, or it should be removed.

Line 239: The term ‘hatching’ is typically reserved for the process of insects emerging from eggs. In this case, “progressing” may be a better word to use.

Line 273: the authors mention that exocuticle formation was inhibited, but, figure 8 implies exocuticle is relatively normal. Whichever is the case, the authors should provide more explanation for their findings here.

When these issues are fixed, this paper would be a great addition to the larger body of literature on ACP.

Author Response

1.       Abstract: Line 23: the authors discuss fifth-instar nymphs exposed to DFB having “the highest mortality.” However, the sentence has no context for what this is compared to. Having read the paper, I believe they mean that mortality was highest in fifth-instar nymphs relative to the other tested life stages. This should be clarified.

Reply: Thanks for reviewer’s valuable and thoughtful suggestions. We have revised the related descriptions, seeing Line 23 to Line 25.

2.       Introduction: Line 83. “Fifth-instar nymphs exposed to DFB further increased mortality.” This sentence provides no affirmation that the authors are repeating already published work. Moreover, the word ‘increased’ has no context, and its use as a verb with ‘nymphs’ as a subject doesn’t really make sense. It isn’t clear to me if the authors mean to imply that they got higher mortality rates that prior work, or that their new research found a better target age. This sentence should be rewritten to clarify these issues.

Reply: Thanks for reviewer’s valuable and thoughtful comments. In this study, spatiotemporal expression profiles analysis revealed that DcCHS has a high expression in the fifth-instar nymph stage. Therefore, we assayed the effect of DFB on Diaphorina citri survival and DcCHS expression level by using fifth-instar nymphs. We considered that fifth-instar nymph stage might be as a better target age for D. citri control. In addition, we have revised the unclear descriptions in previous manuscript, seeing Line 86 to Line 87.   

3.       Line 97: the authors report the use of MEGA 5.0, which is several versions old. If possible, the authors should consider updating their tree with the latest version of MEGA.

Reply: Thanks for your valuable suggestions. We reconstructed the phylogenetic tree by using MEGA 7.0 software, seeing Line 115 to Line 117.

4.       Figure 1A: From an aesthetic point of view, the sequence presentation would look better if presented in more even columns; while preferable, this is not necessary. However, the authors claim to be providing the complete nucleotide sequence, yet all the nucleotide sequence they provide in Figure 1 is translated. If there is no 5’ or 3’ UTR, the authors should state this. If there is UTR, it should be included in Figure 1A. Finally, the authors state in the figure legend that the stop codon of their sequence is TAA, but that actually shown in the image is TAG. This discrepancy should be corrected.

Reply: Thanks for your thoughtful comments. We have revised the incorrect descriptions in previous manuscript, seeing Line 107.

5.       Figure 2: The text in the figure is difficult to read. At the very least, the text should be enlarged. Alternatively, the figure can be made bigger; this might work better if the figure was in box rather than circle format.

Reply: Thanks for your valuable suggestions. We have revised the Figure 2, seeing Line 115 to Line 116.

6.       Figure 3: The Y-axis on each graph is described as "Relative expression level" but this requires a comparison to some baseline. To what are the expression levels "relative"? Do the authors mean "normalized" instead? If the latter, then the Y-axis label should be changed to reflect this reality. For this figure, the authors also provide statistical comparisons to highlight significant differences. However, in each graph, multiple categories are shown, yet, the authors do not clarify, either in the figure legend or in the text, what each category is compared to. This should be explained, since these are cases where the baseline for comparison is intuitive. Alternatively, the authors could use a letter system to indicate categories that are similar.

Reply: Thanks for reviewer’s valuable comments. For tissues expression profiles analysis, we used the midgut as comparison baseline. For developmental stages analysis, we used first-instar nymph stage as comparison baseline. We have revised the Figure 3 using a letter system to indicate categories, seeing Line 138 to Line 139.

7.       Line 144 and following: The authors describe their DFB leaf-dip assay in some detail here and in the Materials and Methods, but what is not clear at any point is whether they exposed their test insects to the DFB for a short period or for the entire 24-48 hrs. This should be made clear in the Materials and Methods, and the language in this paragraph and in the rest of the paper should be altered accordingly. E.g. on line 144, instead of “At 24 h”, the authors could say, “Twenty-four hours after initial exposure” or “After being exposed for 24 h” depending on which method they employed. Additionally, some language implies that insects were re-exposed for the 48-hr test. This should also be clarified. E.g. on line 146, instead of “sharply increased after exposure [to] DFB at 48 h” the authors should say “after a second exposure at 48 h” or “after exposure to DFB for 48 h”.

Reply: Thanks for reviewer’s valuable and thoughtful comments. We have revised the related descriptions in previous manuscript, seeing Line 148 to Line 151.

8.       Line 151: I recommend replacing the word “phenotype” with “morphology”.

Reply: Thanks for your valuable suggestions. We have revised “phenotype” into “morphology”, seeing Line 156.

9.       Line 169-170: the authors describe DcCHS expression as “significantly up-regulated” following RNAi. However, this is the opposite of what would be expected, and is also contradictory to the data presented in Figure 6A. Most likely, this is a misstatement and should be corrected to say “significantly reduced”.

Reply: We are sorry for incorrect descriptions in previous manuscript. We have revised the incorrect descriptions, seeing Line 176.

10.   Line 175 and Figure 7: the authors mention “cumulative molting rate”, but they are not really showing a rate so much as total percent of adults that have molted. The authors would be fine removing the word ‘rate’ from the text and the figure.

Reply: Thanks for reviewer’s suggestions. We have revised the related descriptions, seeing Line 182 to Line 183.

11.   Figure 7: The legend does not discuss the significance values indicated, or how they were calculated. Moreover, the authors do not seem to be making appropriate comparisons. In graphs like this, researchers will typically show how significantly different the Treatment is relative to the Control, not how different one time point is from another. Since the results are cumulative, one would expect that, after sufficient time, enough individuals are affected to create a significant time point, after which, every other time point would be significant. Therefore, making significance comparisons between later time points and the start, as the authors seem to be doing, is meaningless. What is more important is if and when the numbers in the Treatment significantly diverge from the numbers in the Control. The authors should redo their statistical tests to be more in line with this expectation.

Reply: Thanks for your valuable and thoughtful comments. We have revised the Figure 7, seeing Line 202 to Line 203.

12.   Figure 8: The authors do not explain what the blank is. Moreover, it looks very different even from their control (almost like it is a lower magnification). The authors need to explain what this image is meant to show and why it looks different from control. If the image serves no effective purpose, then it should be removed – the control (dsGFP) should be sufficient.

Reply: Thanks for reviewer’s thoughtful comments. We have removed the blank in Figure 8, seeing Line 208 to Line 209.

13.   Line 233-234: this sentence seems internally redundant, and its intended meaning (and role in progressing the authors’ arguments) is not immediately clear. This sentence should be reworded to be clearer, or it should be removed.

Reply: Thanks for reviewer’s thoughtful suggestions. We have deleted the redundant descriptions in previous manuscript, seeing Line 247 to Line 248.

14.   Line 239: The term ‘hatching’ is typically reserved for the process of insects emerging from eggs. In this case, “progressing” may be a better word to use.

Reply: Thanks for reviewer’s valuable suggestions. We have revised “hatching” into “progressing”, seeing Line 254.

15.   Line 273: the authors mention that exocuticle formation was inhibited, but, figure 8 implies exocuticle is relatively normal. Whichever is the case, the authors should provide more explanation for their findings here.

Reply: We are sorry for the incorrect descriptions in previous manuscript. We have revised the relevant descriptions, seeing Line 187 to Line 189 and Line 291 to Line 292